# BmAbl1 Regulates Silk Protein Synthesis via Glutathione Metabolism in *Bombyx mori*

**DOI:** 10.3390/insects13110967

**Published:** 2022-10-22

**Authors:** Sheng Qin, Lingling Sun, Shu Zhang, Xia Sun, Muwang Li

**Affiliations:** 1Jiangsu Key Laboratory of Sericultural Biology and Biotechnology, School of Biotechnology, Jiangsu University of Science and Technology, Zhenjiang 212100, China; 2Key Laboratory of Silkworm and Mulberry Genetic Improvement, Ministry of Agriculture and Rural Affairs, Sericultural Research Institute, Chinese Academy of Agricultural Science, Zhenjiang 212100, China

**Keywords:** *Bombyx mori*, silk yield, ABL tyrosine kinase, protein synthesis, glutathione metabolism

## Abstract

**Simple Summary:**

Silk yield is the primary economic character in sericulture. During the past 5000 years of domestication, great effort has been made to increase cocoon shell weight (CSW). However, silk yield is a complex trait controlled by quantitative trait loci (QTLs). From these QTLs, BmAbl1 tyrosine kinase, which is located at chromosome 1, is considered to be closely related to CSW. In our study, the functional loss of *BmAbl*1 led to altered glutathione metabolism, which ultimately affects the amino acid synthesis and metabolic pathway, impairing silk fibroin protein synthesis. We further investigated the role of *BmAbl*1 in regulating silk fibroin secretion. A new perspective was taken on factors affecting silk fibroin synthesis. This provides a new idea to improve silk fibroin yield. It lays the groundwork for improving silk yield in the future.

**Abstract:**

*Bombyx mori*, domesticated from wild silkworms, is an economic insect that feeds on mulberry leaves and produces silk. In the current study, we demonstrated the contribution of *BmAbl1* in silk protein synthesis. The inhibition and knockout of *BmAbl1* can reduce the larva weight and CSW. The effect on CSW of *BmAbl1* is not on the transcriptional level, but on the translational level. RNA-sequencing data suggested that amino acid synthesis and the metabolism process had a great difference between the *BmAbl1*^-^ and Control strain, particularly glutathione metabolism. An abnormality in glutathione metabolism led to the reduction of free glycine and serine content, which are the main components of fibroin protein. Finally, fibroin protein synthesis has been reduced, including fibroin-heavy chain, fibroin-light chain, and p25 protein. This finding brought to light the role of *BmAbl1* in the silk protein synthesis process.

## 1. Introduction

Silkworms have been domesticated over the past 5000 years from the wild progenitor. To increase the yield of sericulture, one of the main tasks in domestication is to screen a strain with high cocoon quality. The cocoon contains two main structural proteins, fibroin and sericin. Fibroin is the primary composition that accounts for 70–80% of the weight of the silk fiber. Sericin usually accounts for 20–30% of the weight of the silk fiber. Fibroin consists of the heavy chain, light chain, and P25 [1,2]. The molar ratio between these is generally 6:6:1 [3]. The amino acid composition of the heavy chain occupying most of the fibroin is dominated by Gly (46%), Ala (30%), Ser (12%), Tyr (5.3%), and Val (1.8%) [4].

Traditional breeding employs a cross-breeding method based on genetic theory. In recent years, molecular breeding assisted by QTL mapping has provided new technical support for silkworm selection. Numerous researchers have confirmed that a QTL region on chromosome 1 makes a significant contribution to cocoon shell weight (CSW) [5,6,7,8]. Within this QTL region, the tyrosine protein kinase, *BmAbl1*, is considered a potential target gene affecting CSW [9,10].

Abl1, a conserved non-receptor tyrosine kinase, plays key roles in various signaling transform pathways [11]. Many receptors send signals and link extracellular stimuli to other signaling pathways via Abl1, such as Notch, Robo, Eph, and others [12,13,14,15]. The signal has been implicated in the regulation of cytoskeletal rearrangement, axon patterning, cell proliferation, migration, and protein secretion [11,16,17]. The *Drosophila Abl* gene has a great contribution to nervous development, and controls growth cone guidance and synaptogenesis, acting downstream of the Robo receptor [12,18]. In mammals, it is an activity factor of the Ras/Raf/MAPK pathway and PI-3K/Akt pathway regulating the proliferation and differentiation of cells [19]. In *B. mori*, the overexpression of *Ras1* in the posterior silk gland can increase cell size and enhance fibroin production [20,21]. However, whether and how *BmAbl1* contributes to the synthesis of silk protein, and how it works, are still unclear.

In the current research, we inhibited the function of BmAbl1 using a blocking agent and gene knockout, respectively. Compared with the Control group, *BmAbl1*^-^-suppressed individuals showed lighter larva weight and cocoon weight. The expression of silk protein genes was detected by qRT-PCR, but there was no difference between *BmAbl1*^-^ and the Control group. The RNA-sequencing (RNA-Seq) results indicated that amino acid synthesis and the metabolism pathway were affected, especially glutathione metabolism. An abnormality of glutathione metabolism caused glycine (Gly) and serine (Ser) reduction in *BmAbl1^-^*. Ser and Gly are the major components of fibroin protein. The free amino acid test showed that the content of Gly and Ser in the posterior silk gland (PSG) of *BmAbl1^-^* was significantly lower than that of the Control group. Correspondingly, the content of fibroin proteins was decreased in the PSG of *BmAbl1^-^*.

## 2. Materials and Methods

### 2.1. Sample Preparation

*Nistari*, a multivoltine and non-diapausing strain of *Bombyx mori*, was obtained from the Sericultural Research Institute of Chinese Academy of Agricultural Science. The larvae were reared on sufficient fresh mulberry leaves under a constant 12-h light/12-h dark photoperiod at 25 ± 1 °C and 75% ± 3% relative humidity.

### 2.2. Inhibited BmAbl1 Using Dasatinib

Abelson tyrosine kinase was inhibited by Dasatinib (Adooq Bioscience, Nanjing, China). An amount of 1 mg of Dasatinib was dissolved in 205 mL H_2_O to prepare the stock solution. The stock solution was diluted to 0.6 nm and smeared on the fresh mulberry leaves. First-day third-instar larvae of *B. mori* were reared with inhibitor-treated leaves until the wandering stage. The Control group larvae were reared with H_2_O-treated leaves. Each group had three biological replicates. Each replicate contained thirty larvae. All larvae were taken for the same treatment.

### 2.3. Measurement of Tyrosine Kinase Activity

The Protein Tyrosine Kinase (MEIMIAN, Yancheng, China) ELISA Kit was used to measure tyrosine kinase activity. The PSGs of the Treated and Control groups were dissected from the wandering stage of fifth-day fifth-instar larvae. Each group had three biological replicates. Each replicate contained five larvae.

After dissecting the posterior silk gland of the fifth-day fifth-larvae, 100 mg tissues were checked and rapidly frozen with liquid nitrogen. The samples were maintained at 2–8 °C after grinding, and 1 mL PBS (PH7.4) was added. The samples were homogenized by hand, and centrifuged for 20 min at a speed of 2000–3000 rpm to remove the supernatant.

A volume of 40 μL of the sample dilution was added to test the sample well, and then 10 μL of the testing sample was added. HRP-Conjugate reagent (100 μL) was added to each well, except the blank well. It was incubated for 60 min at 37 °C. Then, the liquid was discarded; it was dried by swing; washing buffer was added to every well; it was stilled for 30 s; and then drained; this was repeated 5 times. Chromogen Solution A (50 μL) and B (50 μL) was added, respectively. It was reacted in the dark for 15 min at 37 °C. The stop solution was added, and, finally, absorbance was read at 450 nm after adding the stop solution within 15 min by SpectraMax Paradigm (Molecular Devices, CA, USA).

### 2.4. cDNA Synthesis and Quantitative Real-Time PCR

To validate RNA-Seq, total RNA was extracted from three biological samples of Treated and Control PSGs using RNAiso Plus (TaKaRa, Dalian, China). For each sample, three pairs of PSGs from the Nistari or BmAbl1^-^ were combined. Gel electrophoresis and ultraviolet spectrophotometry were used to determine the integrity and purity of the RNA. One microgram of total RNA from each sample was used to synthesize cDNA using a PrimeScript RT Reagent Kit with gDNA Eraser (Perfect Real Time, TaKaRa)—followed by storage at −20 °C. Real-time qRT-PCR was carried out in a reaction volume of 20 μL, containing 2 μL of template, 10 μL of 2× SYBR Premix EX Taq (TaKaRa), and 0.4 μL of specific primers (10 μM). The PCR amplification efficiency (E) and *R* [2] of each primer pair were calculated from the slope of a standard curve, which was conducted according to MIQE (Minimum Information for Publication of Quantitative Real-Time PCR Experiments) guidelines. The qRT-PCR primer sequences, which were designed based on the consensus sequence of each alignment based on the assembled RNA-Seq, and their efficiencies, are provided in the Appendix A. Real-time qRT-PCR was performed with a Roche lightCycler 96 real-time PCR system, using the following conditions: 95 °C for 5 min, followed by 40 cycles of 95 °C for 10 s, 60 °C for 40 s, and dissociation. The GAPDH gene was used as the reference. Three repeated experiments were set up. The expression of the target gene was calculated by the Ct value, and graphics were created based on these data.

### 2.5. Plasmid Construction

The *BmAbl1*^-^ strain was established based on a binary transgenic CRISPR/Cas9 system. An effector plasmid pXL-[RFP-U6-BmAbl1-sgRNA], which expresses sgRNA targeting *BmAbl1* under the control of the silkworm small nuclear RNA promoter, U6, was constructed through a series of cloning steps. The primers used in the cloning are included in the Appendix A.

### 2.6. Silkworm Germline Transformation

Germline transformation was performed on fresh eggs, within 6 h of egg laying. Thirty positive G0 individuals with red fluorescence were selected from 160 eggs injected with the transgenic system. Preblastoderm embryos were injected with transgenic plasmid mixed with the helper plasmid, and incubated in a humidified chamber at 25 °C for 10–12 days until hatching. The G0 moths were sib-mated or backcrossed with WT moths, and the presence of the selection marker gene in G1 progeny was scored during the late embryonic stage using a fluorescence microscope (Leica M165FC). Heterozygous progeny between IE1-Cas9 and pXL-[IE1-DsRed2-U6-Abl1-sgRNA1 + U6-Abl1-sgRNA2] lines were scored for double fluorescence and used in subsequent experiments.

Genomic DNA was extracted from injected embryos using an Ezup Column Animal Genomic DNA Purification Kit (Sangon, Shanghai, China). Mutagenesis in targeted sites was detected by PCR. The primers that were used in PCR are listed in the Appendix A.

### 2.7. RNA-Sequencing Analysis

There are three biological samples for RNA-Seq. For each sample, three pairs of WSGs were used from the *Nistari* or *BmAbl1^-^* at the third-day fifth-instar larvae. Total RNA was extracted from the whole posterior silk gland dissected from third-day fifth-instar larvae. cDNA libraries were then constructed and sequenced using an Illumina HiSeq2500 platform (Illumina, San Diego, CA, USA). The quality of the raw data was assessed using the FastQC and FastX-Toolkit. Adaptor sequences, unknown sequences (N), and low-quality reads were removed from the raw data. The reference genome sequence of *B. mori* was obtained from SilkDB 3.0 [22]. The clean reads were mapped to the silkworm reference genome using the HISAT2 [23]. The raw counts of each sample were generated by HTSeq [24]. Differential expressed genes were analyzed using the DESeq2 package. *p* values were adjusted for multiple comparisons using the Benjamini–Hochberg method. A gene with adjusted *p* ≤ 0.001 and |log2 Fold Change (Log_2_ FC)| ≥ 1 was considered a DEG. GO enrichment analysis was performed by ClusterProfile [25], and KEGG enrichment analysis was performed by KOBAS 3.0 [26]. A term with *p* ≤ 0.05 was considered an enrichment term.

### 2.8. Phenotypic Measurement of Transgenic Strain

Phenotyped Nistari, BmAbl^-^, and silkworms were fed with ABL inhibitor. Larval weight was measured from first-day fifth-instar to the wandering stage in groups of 30 individuals with three biological replicates. Thirty individuals were weighed together, and then each individual’s weight was averaged. The whole cocoon weight and pupal weight were measured in the same way as above.

### 2.9. Measurement Content of Glutathione (GSH)

A Trace Reducing Glutathione Assay Kit (Jian Cheng, Nanjing, China) was used to measure glutathione (GSH) of the PSG by 5,5′-Dithiobis-(2-nitrobenzoic acid) (DTNB). The samples were extracted from the PSG of the *Nistari* and *BmAbl1^-^* strains at the third-day fifth-instar. Tissues (100 mg) were weighed and added with 9-times PBS (pH = 7.4) for ultrasonic crushing. Centrifugation was conducted at 2500 rpm/min for 10 min. A volume of 0.1 mL of supernatant was taken, and 0.1 mL of precipitant was added. It was centrifuged at 3500 rpm/min for 10 min, and the supernatant was tested at 405 nm using SpectraMax Paradigm (Molecular Devices, San Jose, CA, USA). Every repeat had five individuals’ PSG. Each set of data was repeated three times.

### 2.10. Measurement Content of Free Amino Acids

The samples for measuring the content of free amino acids were obtained from the PSG of the *Nistari* and *BmAbl1^-^* strains at the fifth-day fifth-instar. After the samples were crushed, 1 g of tissue was extracted by adding 50 mL of 0.02 mol/L hydrochloric acid for 30 min. After shaking and filtering, 2 mL of filtrate was added to a centrifuge tube with 2 mL 8% sulfosalicylic acid and left for 15 min. After centrifugation at 10,000× *g* rpm for 10 min, the supernatant was filtered into a 0.45 μm membrane. After sample pretreatment, we performed quantitative analysis using an amino acid analyzer (LA8080, Hitachi, Japan). The sample was pushed through a sulfonic acid cationic resin separation column, and the amino acid mixture was separated into individual amino acids using different PH buffers. The isolated individual amino acids react with ninhydrin reagents to produce purple or yellow compounds. The absorbance at 570 nm and 440 nm was measured for the quantitative analysis of various amino acids. The reaction temperature was 135 ± 5 °C and the injection volume was 20 μL. Every repeat had five individuals’ PSG. Each set of data was repeated three times.

### 2.11. Western Blot

The fibroin heavy chain (Fib-H), fibroin light chain (Fib-L), and P25 protein antibodies (1:4000, ABclonal Technology, Wuhan, China) were used for detecting the expression of Fib-H, Fib-L, and P25 in the PSG of the *Nistari* and *BmAbl1^-^* strains by Western blotting. In addition, all primary antibodies were purified from New Zealand White Rabbits. All primary antibodies we used in this article were sponsored by Professor Tan Anjiang [27], Science of Plant Physiology and Ecology, Chinese Academy of Sciences. Total proteins were extracted from the PSG of the *Nistari* and *BmAbl1^-^* strains. For each sample, there were three pairs of PSGs from the *Nistari* or *BmAbl1^-^* strains. Each 100 mg sample was digested by 1 mL lysis buffer (containing 7 M urea, 2 M thiourea, 40 mg CHAPS, and 200 μL TritonX-100), 10 μL phenylmethanesulfonylfluoride (PMSF), and 0.01 g DTT. The mixture was stewed on ice for 30 min. Protein concentration was determined by a Bradford Protein Assay Kit (Sangon, Shanghai, China). The homogenate was centrifuged at 4 °C and 12,000× *g* for 10 min. We used 12% SDS-PAGE to separate samples at 80 V for 0.5 h and 120 V for 1 h, and then transferred the proteins to PVDF membranes. The tubulin antibody (1:4000, Bioss ANTIBODIES, Beijing, China) was used to level the protein content. The target protein was combined with the corresponding primary antibody. The secondary antibody was HRP Goat Anti-Rabbit IgG (1:5000, ABclonal Technology, Wuhan, China). The tubulin protein was used as the reference.

### 2.12. Statistical Analysis

The statistical differences among the three biological duplicates were determined with ANOVA and Student’s *t*-test by R. Each set of data was repeated three times. Asterisks were used to indicate significant differences (* *p* < 0.05; ** *p* < 0.01; and *** *p* < 0.001).

## 3. Results

### 3.1. BmAbl1 Inhibited by Dasatinib Reduce Larva Weight and CSW

To investigate the function of *BmAbl1*, we inhibited its activity using dasatinib. Dasatinib is a dual-specificity tyrosine kinase inhibitor, and is used as the inhibitor of Abl1 in insects. To test whether dasatinib can be used on *B. mori*, the inhibition efficiency of BmAbl1 activity was measured using the ELISA. The results showed that BmAbl1 activity was significantly inhibited to 41% of the Control (Figure 1A).

We also investigated the difference in larva and cocoon weight between the Treated and Control group. The daily larval weight of the Treated and Control group from the fourth-instar molting stage to the wandering stage showed that the weight of the Treated group was lower than that of the Control at the same time (Figure 1B). The whole cocoon weight (WCW) of the Treated decreased by 5.21% and 8.28% in the female and male individuals than that of the Control group, respectively. The CSW of the Treated group in females and males was, on average, lighter by 18.75% and 18.64% of the Control, respectively. The cocoon layer ratio of the Treated group was reduced by 13.33% and 12.27% in the female and male individuals (Figure 1C). In summary, these results showed that the inhibition of BmAbl1 can reduce larva weight and CSW.

### 3.2. BmAbl1 Knockout Reduces Larva Weight and CSW

To further confirm the effect of *BmAbl1* deficiency on CSW, we constructed a binary transgenic CRISPR/Cas9 system to knockout *BmAbl1* somatically. Two transgenic silkworm lines were established based on a piggyBac transposon-derived vector. One expresses Cas9 ubiquitously under the control of nanos (nos) promoter (nos-Cas9), and the other line expresses the sgRNA targeting *BmAbl1* driven by the silkworm U6 promoter (Figure 2A). After hybridization of the sgRNA and Cas9 strains, 1216 bp-delegation were detected at target sites in offspring (Figure 2B). Two-hundred and eight bases are missing at the mRNA level, including all ninety-seven bases of exon2, and one-hundred and eleven bases of exon3. Frameshift mutation and the early termination of amino acids have taken place at the protein level (Appendix A). To exclude the influence of the inserted element, deletion offspring were crossed with *Nistari* and self-cross. Finally, the homozygous individuals without fluorescence were screened based on PCR and sequencing. The *BmAbl1*^-^ strain showed smaller individuals of the fifth-instar third-day and cocoon size than the normal *Nistari* strain (Figure 2C).

We investigated the weight of larvae of *BmAbl1*^-^ and *Nistari* from the fifth-instar first-day to the sixth day (Figure 3A). The weights of *Nistari* were generally higher than *BmAbl1*^-^ during the whole fifth larva stage. The whole cocoon weight (WCW) of male and female *BmAbl1*^-^ was, on average, lighter by 12.77% and 14.29% of *Nistari*. The CSW of male and female *BmAbl1*^-^ was, on average, lighter by 23.08% and 15.38% of *Nistari*, respectively (Figure 3B).

In order to explore the reason for CSW reduction on a molecular level, we measured the expression of silk fibroin genes in third-day fifth larva. However, we found that the expression of *Fib-H*, *p25*, and *sericin-1* was not significantly changed between *BmAbl1*^-^ and *Nistari* (Figure 3C,D). Only the expression of *Fib-L* gene had a little change (Figure 3C, FoldChange < 0.5).

### 3.3. Glutathione Metabolism and Amino Acid Biosynthesis Were Affected by Knocking Out BmAbl1^-^

To further uncover the influence of knocking out *BmAbl1*, the transcriptome data of third-day fifth-larval PSG was analyzed and compared between *BmAbl1*^-^ and *Nistari*. The RNA-seq raw data were deposited to NCBI SRA with the accession numbers, SRR21798269 to SRR21798274 (https://www.ncbi.nlm.nih.gov/sra, accessed on 4 October 2022). After quality control, clean reads were mapped to the silkworm genome, which was obtained from SilkDB 3.0. In total, 657 differentially expressed genes (DEGs) were identified between *BmAbl1*^-^ and *Nistari* using the DESeq2 package (Appendix A). Compared to *Nistari*, 264 genes were up-regulated and 393 genes were down-regulated in *BmAbl1*^-^ (Figure 4A).

We employed the related functions of DEGs using enrichment analysis (Appendix A). GO enrichment analysis showed that glutathione and tyrosine metabolic processes were significantly affected in *BmAbl1^-^* (Figure 4B). The molecular functions of DEGs were primarily involved in glycosyltransferase activity, cation binding, phospholipid binding, glutathione transferase activity, and pentosyltransferase activity.

KEGG enrichment analysis indicated that amino acid biosynthesis and metabolism pathways had a significant difference between *BmAbl1*^-^ and *Nistari*, especially glutathione metabolism; glycine, serine, and threonine metabolism; phenylalanine metabolism; and tyrosine metabolism (Figure 4C, Appendix A). The DEGs in tyrosine metabolism and phenylalanine metabolism pathways were mainly up-regulated in *BmAbl1*^-^, but were down-regulated in other pathways.

Both GO and KEGG enrichment analyses implicated that the functional defect of *BmAbl1* inhibited the glutathione metabolism process. Glutathione, as an antioxidant, widely exists in animals, plants, and fungi. It can be resolved into glutamate, cysteine, and glycine. This process involves glutathione transferase (EC:2.5.1.18), glutathione gamma-glutamate hydrolase (EC:3.4.19.13), gamma-glutamyltransferase (EC:2.3.2.2), membrane alanyl aminopeptidase (EC:3.4.11.2), and Pep (Appendix A). In *B. mori*, glutathione transferase (EC:2.5.1.18) involves BMSK0012237, BMSK0003439, BMSK0003598, BMSK0003983, and BMSK0003984. We observed that the expression of these genes was significantly down-regulated in *BmAbl1^-^*, except BMSK0003439. The qRT-PCR results demonstrated the RNA-seq results (Figure 5A).

It is particularly noteworthy that glutathione transferase Delta2 (BMSK0003598), one of the subunits of glutathione transferase (EC:2.5.1.18) in *B. mori*, is highly expressed in the glutathione metabolism process of *Nistari*, but is extremely suppressed in the *BmAbl1*^-^ strain. This result implied that glutathione metabolism was slowed, and less Gly was produced in the silk gland. To detect whether it specifically expressed in the silk gland, we compared the expression level of glutathione transferase Delta2 in different organs between the *Nistari* and *BmAbl1*^-^ strain. We found that glutathione transferase Delta2 had the second-highest expression level in the posterior silk gland of *Nistari*. The deletion of *BmAbl1* resulted in a significant decrease in glutathione transferase Delta2 expression in posterior silk glands, but no change or increase in other tissues (Figure 5B). We also observed that the expression of pyrimidodiazepine synthase (BMSK0006405, EC:1.5.4.1) and glucose-6-phosphate dehydrogenase (BMSK0001869, EC:1.1.1.49) was significantly inhibited in *BmAbl1^-^*, and it will slow the conversion between reduced glutathione (GSH) and oxidized glutathione (GSSG).

The reduction of Gly content might further cause the reduction of Ser and threonine (Thr) content. KEGG enrichment results also showed that the glycine, serine, and threonine metabolism pathways changed in *BmAbl1*^-^. We verified the expression of DEGs in the glycine, serine, and threonine metabolism pathways using qRT-PCR. The change of threonine ammonia-lyase (BMSK0014693, EC:4.3.1.19), related to serine metabolism, had a very low expression level and could not be detected. This suggested that the changes in Ser content might not be caused by threonine ammonia-lyase (Appendix A).

According to pathway enrichment analysis, a series of enzymes affecting the amino acid synthesis and metabolism were also identified from DEGs. We performed qRT-PCR to verify the expression of these DEGs. The results were consistent with RNA-seq data. Compared to *Nistari*, the expression of fumarylacetoacetase in *BmAbl1^-^* (BMSK0001189, EC:3.7.1.2) was significantly increased in the tyrosine metabolism pathway (Figure 5C). The expression of pyrroline-5-carboxylate reductase (BMSK0003392, EC:1.5.1.2), ribulose-phosphate 3-epimerase (BMSK0005067, EC:5.1.3.1), argininosuccinate lyase (BMSK0005586, EC:4.3.2.1), and aminoacylase-1 (BMSK0007210, EC:3.5.1.14) were significantly decreased in the biosynthesis of the amino acids pathway (Figure 5D). Changes in these enzymes might also alter the contents of other amino acids.

### 3.4. Abnormal Glutathione Metabolism Reduced Glycine and Serine Content in PSG of BmAbl1^-^

RNA-seq and qRT-PCR results implicated that diminished glutathione metabolism might cause the accumulation of glutathione, and decrease free Gly and Ser content (Appendix A). To confirm our hypothesis, we first compared the glutathione content in the PSG of fifth-day fifth-instar larvae between *BmAbl1^-^* and *Nistari*. The ELISA experiment results showed that the glutathione content in *BmAbl1^-^* (avg = 78.50 μmol/mg) significantly increased to 1.49 times more than *Nistari* (avg. = 52.55 μmol/mg) (Figure 6A, *p* = 0.0002). This result demonstrated that abnormal glutathione metabolism caused by knocking out *BmAbl1^-^* led to glutathione accumulation.

Fibroin is composed of Fib-H, Fib-L, and p25, with a 6:6:1 molar ratio. The amino acid composition of Fib-H is Gly (46%), Ala (30%), Ser (12%), Tyr (5.3%), and Val (1.8%) [4]. The key enzymes (such as EC:2.5.1.18, EC:3.4.19.13, and EC:2.3.2.2) in the glutathione decomposition pathway were significantly down-regulated in *B**m**A**bl^-^*. Abnormalities of the glutathione metabolism pathway may induce the decrease of free Gly and Ser (Appendix A). To further confirm our hypothesis, we next measured free amino acid contents in the WSG of fifth-day fifth larvae. The results showed that Gly and Ser had a significant down-regulation in the *BmAbl1*^-^ strain, but Ala had no difference (Figure 6B).

We further compared three major fibroin proteins content in the PSG of fifth-instar larvae between *BmAbl1*^-^ and *Nistari*. The western blot results showed that Fib-H, Fib-L, and P25 protein were all down-regulated compared to *Nistari* (Figure 6C).

## 4. Discussion

Since the silkworm was domesticated as an economic insect 5000 years ago, the cocoon quantity is always an important trait in artificial screening. However, a series of factors can affect the cocoon quantity of a silk strain, such as the development of the silk gland [21], the number of silk gland cells [28], the capabilities of DNA replication [29], and protein synthesis [20,30]. On the molecular level, energy supply, the content of free amino acid, mRNA content of fibroin, the number of ribosomes, and the coordination of the regulatory system are all affecting factors of silk protein synthesis in the silk gland of fifth-instar larvae. Many efforts have been made to find out the major genes which are strongly related to silk yield [5,6,7,8]. Chromosome 1 is widely confirmed to contribute to cocoon shell weight in previous work. *BmAbl1* is one of the potential targets in candidate regions. A recent report suggested that *BmAbl1* had a strong correlation with cocoon shell weight [9,10]. However, how *BmAbl1* affects the synthesis of silk protein is still unknown.

In the current study, our results suggested that the loss-of-function of *BmAbl1* had little effect on the expression of fibroin and sericin1 genes, but the synthesis of fibroin protein was reduced significantly. RNA-Seq data suggested that the glutathione metabolism pathway was suppressed through the down-regulation of glutathione-S-transferase (GST) and gamma-glutamyltransferase (GGT) in *BmAbl1*^-^. The free amino acid measurement results showed that the inhibition of glutathione metabolism led to the reduction of free Gly and Ser, which are the major components of fibroin protein.

Some questions should be answered to address how defective *BmAbl1* down-regulated the expression of GST and GGT (Figure 7). The first question is which transcription factor (TF) regulates by *BmAbl1*. A potential TF is BMSK0005992, a homologous of *cnc* in *D. melanogaster* and *Nrf2* in mice. It has been demonstrated that Nrf2 is an essential activator in the expression of GST family genes in mice [31,32]. ChIP-sequencing data also supported that *GstD2* and *GstE6* were the targets of CNC in *D. melanogaster* [33]. Misra et al. have demonstrated that CNC causes changes in the expression of enzymes, including 36 different P450, 17 GSTs [34]. This is consistent with our observation in RNA-seq data. The next question is how Abl1 activates the transcription factor activity of CNC. It has been known that c-Abl phosphorylated protein kinase C (PKC) can trigger Nrf2 nuclear translocation [35,36]. A previous study suggested that c-Abl is activated by PKC and c-Abl phosphorylates, and thereby further activates PKC [35]. The phosphorylation of Nrf2 by PKC promoted its dissociation from Keap1, and translocated into the nucleus to activate ARE-mediated gene expression [32,36]. However, this potential pathway needs to be further verified in *B. mori*.

## 5. Conclusions

In this work, we confirmed the contribution in the synthesis of silk protein using chemical inhibitor and knockout BmAbl1 methods. The functional defect in BmAbl1 can reduce the weight of the larva and the cocoon. Our results suggested that the loss of function of BmAbl1 impaired the synthesis of fibroin protein via glutathione metabolism. The inhibition of glutathione metabolism reduced the content of free Gly and Ser in the *BmAbl1^-^* strain. As the major component of fibroin protein, the reduction of free Gly and Ser content further affected silk protein production.

## Figures and Tables

**Figure 1 insects-13-00967-f001:**
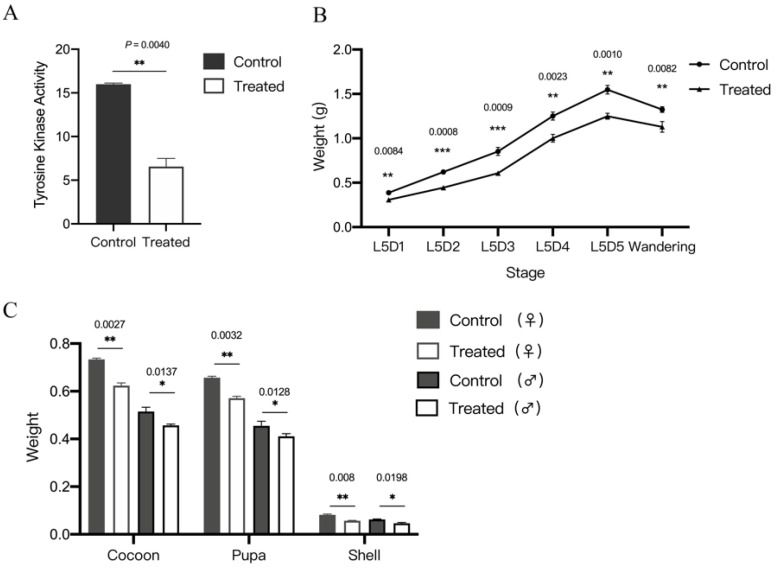
Inhibited BmAbl1 led to a reduction of cocoon shell weight (CSW). (**A**) Dasatinib inhibited the activity of BmAbl1; (**B**) inhibited BmAbl1 led to larva weight being lighter than Control group during the fifth instar; (**C**) inhibited BmAbl1 led to cocoon weight, pupa weight, and CSW being lighter than Control group during the fifth instar. Standard deviation was calculated by all biological replicates. * *p* < 0.01, ** *p* < 0.005, *** *p* < 0.001, *t*-test, n = 30 for Control and Treated, respectively.

**Figure 2 insects-13-00967-f002:**
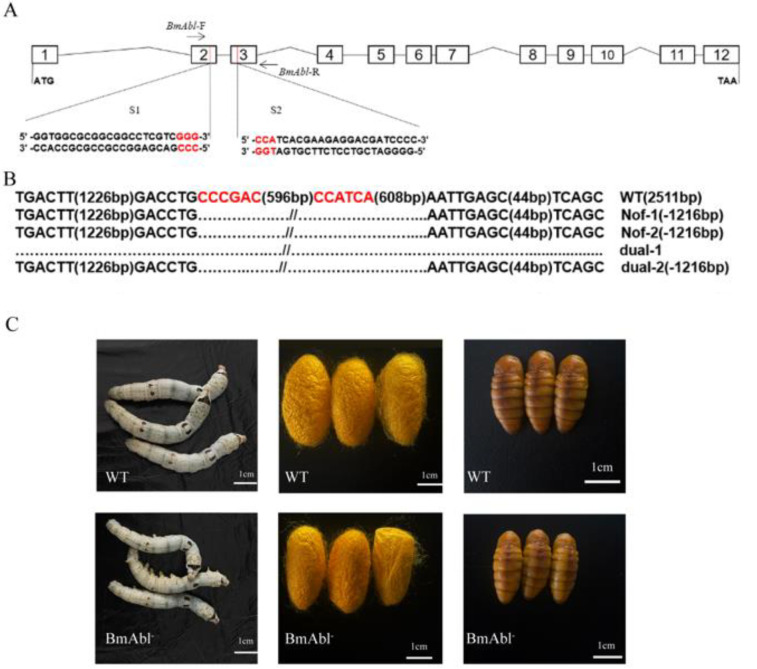
Target site design of CRISPR/Cas9 and verification. (**A**) Two target sites were designed on *BmAbl1*. One is on the second exon; another is on the third exon; (**B**) sequencing results show that the fragment has been deleted between two target sites; WT indicates *Nistari* strain. Dual-1 and dual-2 are *BmAbl1* defective strains based on CRISPR/Cas9 method. Nof-1 and nof-2 are *BmAbl1* defective strains that removed fluorescence protein from the genome through cross-fertilizing with *Nistari* and self-fertilizing. (**C**) Developmental change of bombyx larvae and pupa between WT and BmAbl^-^. *BmAbl1*^-^ strain showed smaller individual (average 0.73 g) and cocoon size (average 0.96 g) than wide type (average 1.00 g, 1.15 g, respectively) (scale bar, 1 cm.).

**Figure 3 insects-13-00967-f003:**
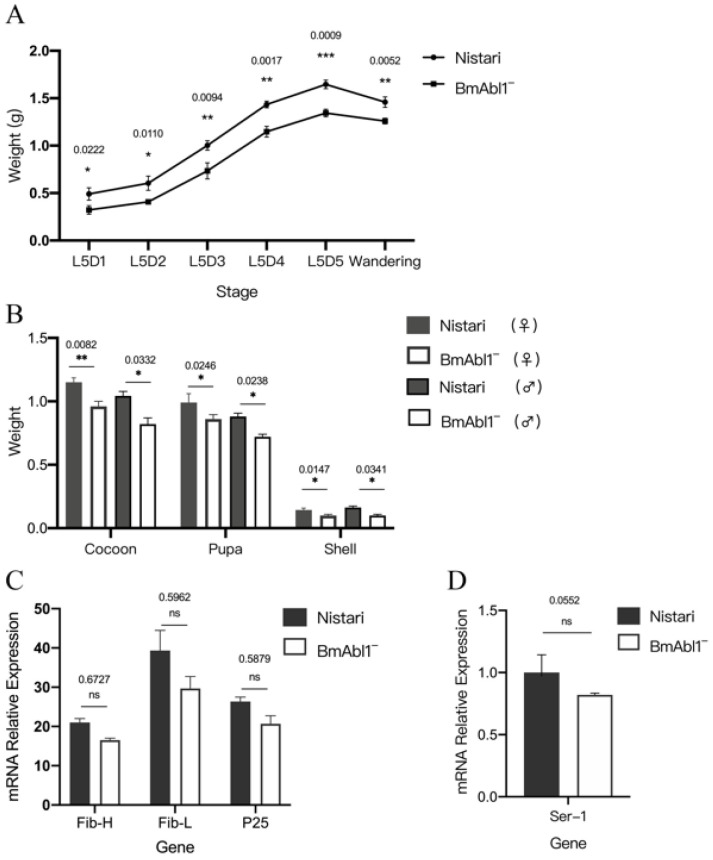
Knockout *BmAbl1* led to a reduction of cocoon shell weight (CSW), but not at the transcription level. (**A**) Larva weight was decreased during the fifth-instar larva of *BmAbl1*^-^ strain compared to Nistari; (**B**) cocoon weight, pupa weight, and CSW were decreased in *BmAbl1*^-^ strain compared to Nistari, both male and female; (**C**) the expression levels of fibroin heavy-chain, fibroin light-chain, and p25 gene were not different between *BmAbl1*^-^ and *Nistari*; (**D**) the expression level of *sericin 1* was not different between *BmAbl1*^-^ and *Nistari*. Standard deviation was calculated by all biological replicates. * *p* < 0.01, ** *p* < 0.005, *** *p* < 0.001, *t*-test, n = 30 for Control and n = 30 for Treated; each group had three biological replicates.

**Figure 4 insects-13-00967-f004:**
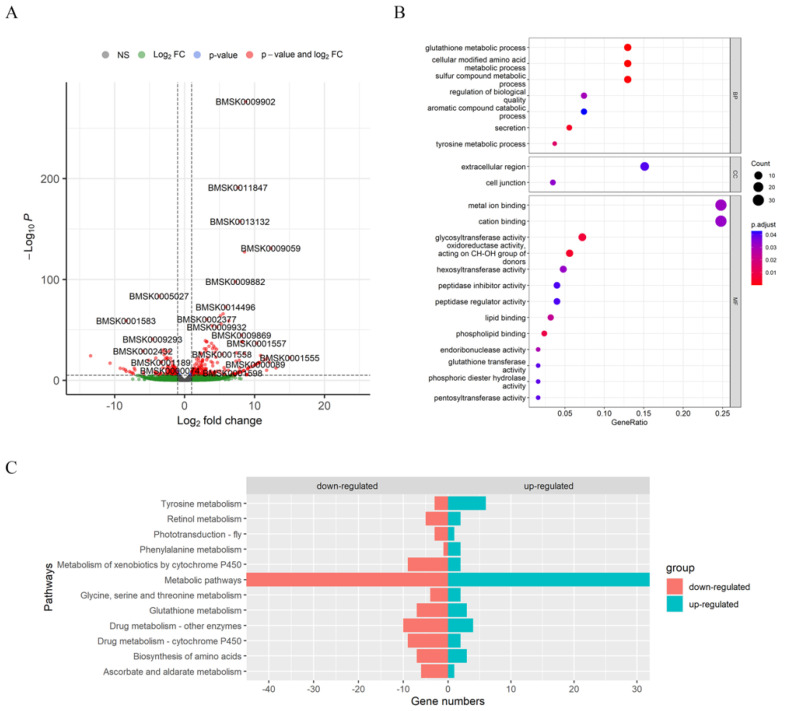
RNA-sequencing data uncovered differences between *BmAbl1^-^* and *Nistari* in transcription level. (**A**) The volcano plot showed 657 differentially expressed genes (DEGs) between *BmAbl1*^-^ and *Nistari*. Compared to *Nistari*, 264 genes were up-regulated and 393 genes were down-regulated in *BmAbl1*^-^. Red dots indicate DEGs, and green dots indicate non-DEGs; (**B**) GO enrichment analysis result; (**C**) KEGG enrichment analysis results. Blue bars indicate the up-regulated genes in enriched terms. Red bars indicate the down-regulated genes in enriched terms.

**Figure 5 insects-13-00967-f005:**
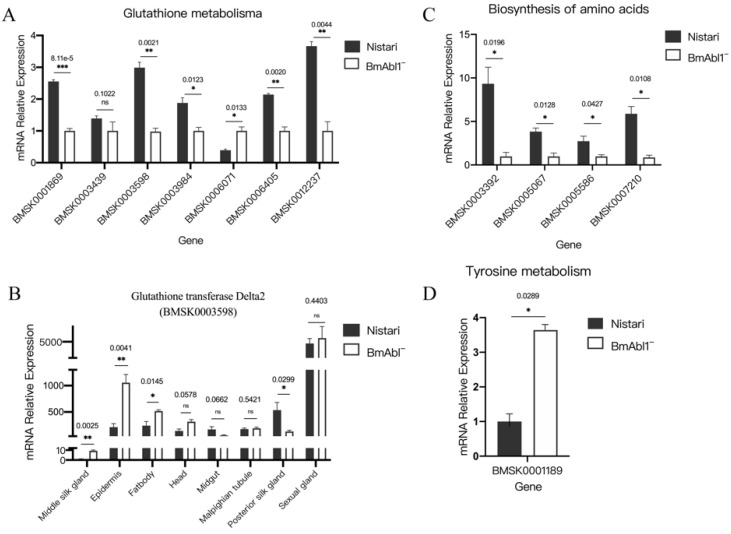
Verification of DEGs expression by qRT-PCR. (**A**) Verification of DEGs in glutathione metabolism pathway; (**B**) the expression of glutathione transferase Delta2 in each tissue was specifically decreased in the posterior silk gland; (**C**) verification of DEGs in tyrosine metabolism pathway; (**D**) verification of DEGs in the biosynthesis of amino acids pathway. Standard deviation was calculated by all biological replicates. * *p* < 0.01, ** *p* < 0.005, *** *p* < 0.001, *t* test, each group had three biological replicates.

**Figure 6 insects-13-00967-f006:**
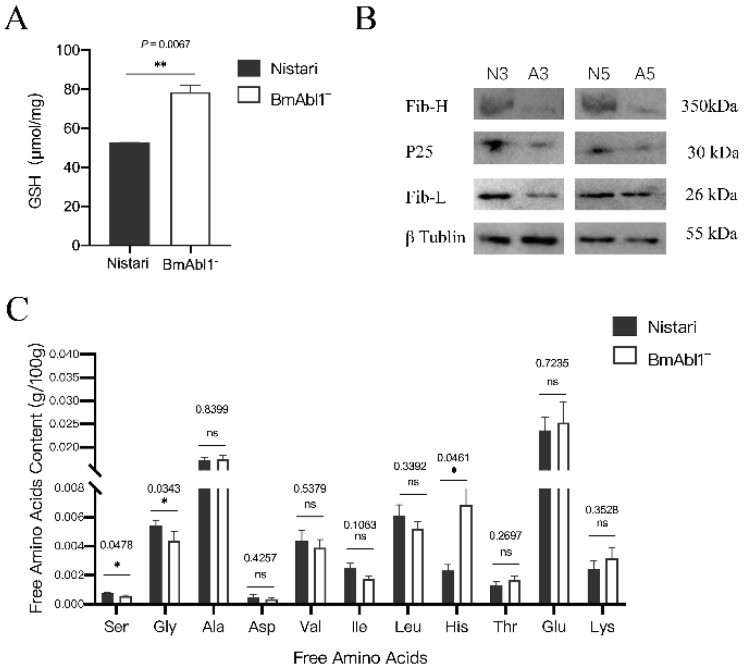
Abnormality of glutathione metabolism led to the reduction of fibroin protein. (**A**) Inhibition of glutathione metabolism pathway led to the accumulation of glutathione; (**B**) inhibition of glutathione metabolism pathway led to reduction of Gly and Ser; (**C**) the content of fibroin proteins decreased in BmAbl1^-^. N3: Nistari l5d3; A3: BmAbl^-^l5d3; N5: Nistari l5d5; A5: BmAbl^-^l5d5. Standard deviation was calculated by all biological replicates. * *p* < 0.01, ** *p* < 0.005, *t*-test.

**Figure 7 insects-13-00967-f007:**
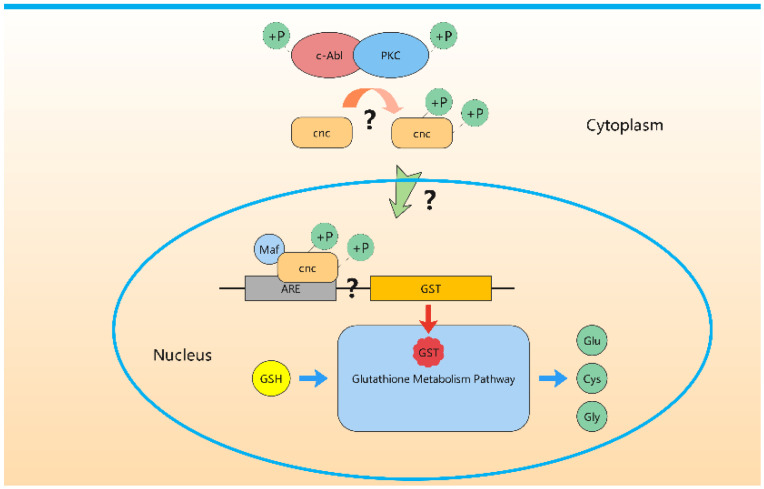
Potential pathway of GST activated by Abl1 to degrade GSH into Gly. “?” indicates the inferred pathway from the literature.

## Data Availability

The data presented in this study are available in Appendix A. The RNA-seq raw data were deposited to NCBI SRA with the accession numbers, SRR21798269 to SRR21798274 (https://www.ncbi.nlm.nih.gov/sra, accessed on 4 October 2022).

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
