# Peer review of "BmAbl1 Regulates Silk Protein Synthesis via Glutathione Metabolism in Bombyx mori"

_insects, 2022, doi:10.3390/insects13110967_

Round 1

Author Response

Response:

Thank you for your comments concerning our manuscript. Those comments are all valuable and very helpful for revising and improving our paper. We have studied comments carefully and have made a revision which we hope meet with approval. The main corrections in the paper and responses to the comments are listed in the attached PDF file.

Reviewer 2 Report

In this paper, the BmAbl was knocked out to study the function on the synthesis of silk protein. The experimental results showed that the mRNA level after the gene was knocked out had little effect on the expression of silk protein gene, but it did have a significant effect on the amount of cocoon silk. Therefore, the transcriptomics method was used to explore the reason, and it was found that their amino acid synthesis and metabolism were significantly different. The authors believe that the abnormal metabolism of glutathione leads to the decrease of free glycine and serine content, thus affecting the synthesis of silk protein. The research results in this paper are rich and provide a new perspective on silk protein synthesis. However, some experimental evidence in this paper needs to be further supplemented and improved before it can be published.

1.      According to the previous literature, BmAbl gene is a gene that has undergone artificial selection, and its expression amount is higher in high silk content cocoon than that of low silk content silk varieties. Why did the author choose to knock it out rather than over express it to study the effect of this gene on silk protein synthesis? This is closer to the author's goal of increasing silk production. Because the gene is highly expressed in the silk gland and should play an important role in the synthesis of silk protein. Knockout will naturally affect the synthesis of silk protein.

2.     For knockout individuals, the author believes that homozygous knockout individuals have been obtained, but the molecular detection evidence of knockout individuals is insufficient. The author only provided a genome knockout sequence map. How about the sequence at the mRNA level? How many bases are missing at the mRNA level, and what changes have taken place in the protein level? Frameshift mutation, early termination or partial deletion of amino acids? This information is extremely important for the later phenotype and the function of this protein. The author should supplement these results.

3.     Since the author knocks out BmAbl, how about the expression of BmAbl gene in the knockout individuals? It is better use qRT-PCR to detect it, and if possible, use WB to detect and quantify it at the protein level to prove that the homozygous mutation is an effective mutation.

4.     The author has performed transcriptome analysis. Please provide the registration number of the sequencing data in NCBI.

5.     The author detected the silk fibroin in PSG with the antibodies after knockout at the protein level, and found that silk fibroin decreased significantly. For the antibody used in this article, the author does not clearly describe in the method part. Whether the antibodies were purchased or prepared. If they are prepared, important information such as whether the antigen used is polypeptide or whole protein should be provided. Also, how specific is the antibody? Please provide the information of antibody specificity test.

Author Response

Thank you for your comments concerning our manuscript. Those comments are all valuable and very helpful for revising and improving our paper. We have studied comments carefully and have made a revision which we hope meet with approval. The main corrections in the paper and responses to the comments are listed in the attached PDF file.

Round 2

Reviewer 2 Report

In the revised version, the author had replied the comments one by one and revised to the  suggestions mentioned last time. In the feedback draft, I had some doubts about the detection results of silk fibroin heavy chain protein in the WB  of silk fibroin antibody provided by the author. The author did not mark the protein marker, only marked 350 kD at the top band. What is the molecular weight of the strong signal band below? Are they possible signals of silk fibroin heavy chain protein? Whether it is possible that the top sample may be caused by too much sample loading, and some of it may remain in the glue hole, please think about the result further.

Author Response

Dear Reviewer:

Thanks very much for taking your time to review this manuscript. I really appreciate all your comments and suggestions! Our detail response is listed in the PDF file of attachment. We hope this revision should answer your question.
